# SSProNet: Secondary Structure aware Graph Neural Network for Protein Representation Learning

## Abstract

Graph structures are widely leveraged to represent proteins. However, to a large extent, proteins fold into complex three-dimensional conformations that cannot be entirely well-captured by graphs built only from sequence adjacency or distance cutoffs. In this paper, we discover that a more faithful characterization comes from secondary structure elements—such as $\alpha$-helices and $\beta$-sheets—that reflect recurring local motifs and stabilizing hydrogen-bond patterns. To this end, we propose a new graph neural network framework that augments node representations with the secondary structure assignment of each residue and introduces a novel edge-construction strategy based on hydrogen bonds weighted by their energetic strength. This formulation captures both local structural context and long-range couplings essential to protein stability. On commonly used benchmarks, our model achieves the leading accuracy compared with state-of-the-art methods while providing improved interpretability through biologically meaningful edges. These results highlight the promise of secondary-structure-aware, energy-weighted graphs as an effective inductive bias for protein representation learning.

## 1 Introduction

Graph Neural Networks (GNNs) have emerged as powerful learning paradigms for complex, relational data, with successes on social networks (Easley & Kleinberg, 2010), knowledge graphs (Easley & Kleinberg, 2010), molecular graphs (Wu et al., 2018), and biological networks (Barabasi & Oltvai, 2004), as well as for modeling 3D objects (Simonovsky & Komodakis, 2017), manifolds (Bronstein et al., 2017), and source code (Allamanis et al., 2017). Benchmarks such as the Open Graph Benchmark (OGB) have further catalyzed progress by standardizing tasks and evaluation (Hu et al., 2020).

**Proteins as graphs.** Proteins are composed of amino acids and realize diverse cellular functions by folding into three-dimensional (3D) conformations. Beyond the one-dimensional (1D) peptide sequence, each residue has 3D coordinates in space; effective modeling must therefore leverage both views. Notably, proteins with similar sequences can adopt very different folds (Alexander et al., 2009), whereas proteins with similar folds may have entirely different sequences (Agrawal & Kishan, 2001). These observations motivate representation learning methods that couple 1D sequence and 3D structure (Liu et al., 2022; Fout et al., 2017; Jumper et al., 2021; Gao et al., 2021; Gao & Ji, 2019; Yan et al., 2022; Wang et al., 2022b; Yu et al., 2022; Xie et al., 2022; Gui et al., 2022; Luo et al., 2022; Baldassarre et al., 2021; Jing et al., 2020; Zhang et al., 2022; Hermosilla & Ropinski, 2022; Fan et al., 2022; Hu et al., 2024).

**From proximity proxies to biophysical edges.** Radius cutoffs and sequence windows are convenient, but do residues "interact" merely because they are close in space or adjacent in sequence, or because specific chemical and geometric conditions are satisfied (e.g., donor/acceptor compatibility and orientation)? If proximity were the right criterion, why would model performance hinge so strongly on brittle hyperparameters (window size, cutoff radius) instead of stable, mechanistic rules? Prior work improves parts of this picture—CDConv separates discrete sequence from continuous geometry (Fan et al., 2022), ProNet enforces hierarchical completeness (Wang et al., 2022a),

CoupleNet couples dual graphs (Hu et al., 2024), and SCHull offers a sparse, connected scaffold (Wang et al., 2025)—yet the edge decision itself often remains a proximity proxy.

Protein structure is organized into *secondary-structure* elements (e.g., $\alpha$-helices, $\beta$-sheets) stabilized primarily by hydrogen bonds. As outlined in Schulz & Schirmer (2013), organization spans primary, secondary, tertiary, and quaternary levels and extends to supersecondary motifs and domains (Fig. 1a). In practice, tools such as DSSP (Hekkelman et al., 2025) provide residue-level secondary-structure assignments and identify backbone hydrogen bonds (examples in Fig. 1b). These annotations suggest a more faithful inductive bias: *nodes* should encode secondary-structure context, and *edges* should reflect stabilizing interactions—with strengths that vary—rather than distance alone. This is precisely the gap we target next with SSProNet.

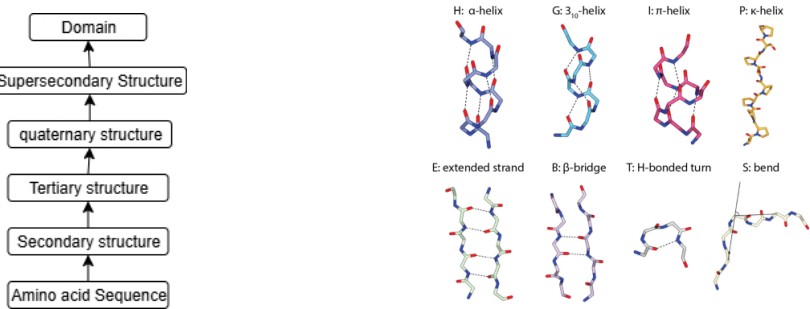

(a) Levels of structural organization in proteins.  (b) Examples of secondary structure elements.

Figure 1: (a) Protein structure levels, from primary to quaternary (plus supersecondary and domains). (b) Common secondary-structure elements such as $\alpha$-helices, $\beta$-sheets, and turns, from DSSP (Hekkelman et al., 2025).

### 1.1 CONTRIBUTION

We introduce *SSProNet*, a GNN that (i) enriches node features with each residue's *secondary-structure assignment* and (ii) defines *edges* via *backbone hydrogen bonds* weighted by their *energetic strength*. Messages flow on this energy-weighted H-bond graph, fused with proximity edges that come from radial graph construction. To ensure geometric robustness, we adopt the SE(3)-invariant descriptors from ProNet (Wang et al., 2022a), derived from local residue frames and inter-residue geometry, which guarantee a complete and rotation/translation-invariant structural representation. Architecturally, SSProNet is compatible with dual-stream coupling ideas (as in Hu et al. (2024)) yet replaces "who-talks-to-whom" with a biophysically grounded criterion; it is complementary to CDConv's separation of discrete/continuous displacements (Fan et al., 2022), and can be paired with SCHull when a provably sparse/connected scaffold is desired (Wang et al., 2025).

### 1.2 PAPER OUTLINE

Section 2 provides the necessary background for GNN-based protein representation learning and the motivation for our SSProNet. Section 3 formally introduces SSProNet, especially detailing its graph construction (secondary-structure nodes and energy-weighted H-bond edges) and invariant features. Section 4 presents experiments, including performance comparisons against the state-of-the-art SCHull framework (Wang et al., 2025). Section 5 concludes the paper.

## 2 BACKGROUND AND MOTIVATION

### 2.1 PRELIMINARY

We model a protein as a 3D graph $G = (V, E, \mathbf{P})$, where $V$ indexes residues (or atoms), $E$ is an edge set, and $\mathbf{P} = \{\mathbf{P}_i \in \mathbb{R}^3\}_{i \in V}$ are coordinates (by default $C_\alpha$ for residue graphs). A representation $\Phi(G)$ is *SE(3)-invariant* if $\Phi(R\mathbf{P} + t) = \Phi(\mathbf{P})$ for any rotation $R \in \mathrm{SO}(3)$ and translation $t \in \mathbb{R}^3$; it is *complete* (up to rigid motion) if $\Phi(G) = \Phi(G')$ implies the coordinates of $G'$ are

congruent to $G$. Across biomolecular GNNs, four properties consistently drive performance and robustness: (i) SE(3) symmetry handling (invariance/equivariance), (ii) *completeness*/expressivity of geometric encodings, (iii) a graph topology that is sparse, connected, and maximally informative, (iv) *biologically grounded* priors (e.g., secondary structure, hydrogen bonds).

## 2.2 Hierarchical SE(3)-aware encoders

**Coarse-to-fine structure.** ProNet (Wang et al., 2022a) introduced hierarchical encoders that build SE(3)-invariant, provably *complete* descriptors at three levels: (1) **amino-acid** (residue) with local frames and inter-residue geometry; (2) **backbone** augmenting with plane/dihedral relations to disambiguate chain orientation; (3) **all-atom** incorporating side-chain torsions for fine-grained distinction. Interaction blocks (Hier-Geom-MP) integrate these descriptors into edge-gated message passing with residual updates and invariant graph readout. This architecture preserves SE(3) symmetry while maintaining discriminative power across scales, and serves as the backbone we inherit.

## 2.3 Coupling sequence and 3D geometry

**Continuous–discrete fusion.** Protein neighborhoods have two distinct regularities: 1D sequence (regular, discrete) and 3D space (irregular, continuous). CDConv (Fan et al., 2022) addresses this by convolving over a hybrid neighborhood (continuous displacements $\delta$ and discrete sequence offsets $\Delta$) with offset-specific kernels, thereby reducing interference between the two modalities while letting them interact.

**Two-graph coupling.** CoupleNet (Hu et al., 2024) operationalizes the idea with two explicit edge families—*sequence* (small $|\Delta|$) and *radius* ($\|\mathbf{P}_i - \mathbf{P}_j\| \leq r$)—and performs coupled node–edge updates. After pooling, it expands spatial thresholds to grow the receptive field as features become coarser. The key takeaway is that architectural separation of sequence and spatial relations simplifies learning and stabilizes training.

## 2.4 Graph construction paradigms

**Radius/$\varepsilon$-graphs and kNN.** Cutoff ($\varepsilon$) or kNN graphs are ubiquitous for coverage and simplicity, but can be either overly dense (hurting sample efficiency) or fragile (hurting connectivity), and may admit geometric ambiguities (distinct structures sharing similar local neighborhoods).

**Rigid, sparse, connected alternatives.** Recent *rigidity-aware* constructions (e.g., spherical-convex-hull or related projections) (Wang et al., 2025) aim for graphs with theoretical guarantees: low edge density, connectivity, and improved identifiability when paired with metric/dihedral edge attributes. These designs reduce spurious edges yet keep enough structure to reconstruct geometry up to isometry, improving downstream stability.

## 2.5 Biology-grounded priors

**Secondary structure and solvent accessibility.** DSSP (Hekkelman et al., 2025) remains the reference for assigning per-residue secondary structure (H/E/C/... variants) and solvent accessibility from PDB coordinates. These labels summarize recurring local conformations (helices, sheets, loops) and exposure, providing interpretable priors that complement purely geometric channels.

**Hydrogen bonds (H-bonds).** DSSP identifies backbone hydrogen bonds using an electrostatics-based energy criterion rooted in Kabsch–Sander. More negative energies indicate stronger bonds; common practice keeps only stabilizing bonds (e.g., threshold $h < 0$ kcal/mol, such as $-0.5$). Importantly, H-bonds are *nonlocal* along sequence and can bridge distant 3D regions (inter-strand $\beta$ ladders, helix capping, long-range turns). As graph edges, they add sparse, physically interpretable couplings that typical radius graphs miss.

## 2.6 Positioning our approach

The above motivates two design decisions we adopt in the below Section 3:

- **Keep the encoder, change the graph.** We retain ProNet's hierarchical, SE(3)-invariant message passing (*capacity held constant*) and instead *redefine the topology* to include energy-filtered H-bond edges on top of a light proximity scaffold. This isolates the effect of biology-grounded connectivity.
- **Add lightweight residue priors.** We inject DSSP-derived secondary-structure and solvent-accessibility channels—interpretable cues that bias the model toward known structural regularities with minimal parameter overhead.

In contrast to prior two-graph schemes (sequence+radius) (Hu et al., 2024) or continuous–discrete kernels (Fan et al., 2022), our hybrid edge set introduces *chemistry-anchored* long-range constraints (H-bonds) while preserving the simplicity and coverage of a radius scaffold. Combined with complete hierarchical encoders (Wang et al., 2022a), this yields message passing over edges that are both *geometrically* informative and *biophysically* meaningful.

## 3 SSProNet: Secondary Structure aware Graph Neural Network for Protein Representation Learning

SSProNet builds on ProNet's (Wang et al., 2022a) hierarchical, SE(3)-invariant encoders while introducing a biology-grounded graph and residue priors. Message passing operates on a hybrid edge set that combines generic proximity contacts with hydrogen-bond couplings anchored in protein chemistry.

### 3.1 Graph construction

We represent each protein chain as a residue graph $G = (V, E)$ with one node per residue and $C_\alpha$ coordinates $\mathbf{P}_i \in \mathbb{R}^3$. The edge set is the union of a generic proximity graph and a biology-grounded hydrogen-bond graph: $E = (E_{\mathrm{rad}} \cup E_{\mathrm{HB}}) \setminus \{(i, i) \mid i \in V\}$.

**Radius (proximity) edges.** We connect residues that are spatially close:
$$E_{\mathrm{rad}} = \{(i, j) : \|\mathbf{P}_i - \mathbf{P}_j\|_2 \leq r\},$$
with a cutoff $r$ (default 10 Å) and an optional degree cap to bound neighborhood size. This provides a light, connected scaffold capturing generic short-range contacts.

**Hydrogen-bond edges.** From backbone hydrogen bonds identified by a standard secondary-structure tool (e.g., DSSP (Hekkelman et al., 2025)), we form directed edges between reported donor–acceptor residue pairs. Let $E_{ij}$ denote the associated H-bond energy (more negative indicates stronger bonding). We retain only stabilizing bonds,
$$E_{\mathrm{HB}} = \{(i, j) : \text{H-bond reported between } i \text{ and } j \text{ and } E_{ij} \leq h\},$$
with threshold $h < 0$ (default $h = -0.5$ kcal/mol). Unless stated otherwise, energies are used for *filtering* (to control precision/recall of $E_{\mathrm{HB}}$) rather than as per-edge weights; undirected variants symmetrize by adding $(j, i)$ whenever $(i, j) \in E_{\mathrm{HB}}$.

**Rationale.** $E_{\mathrm{rad}}$ supplies coverage and local connectivity, while $E_{\mathrm{HB}}$ injects a sparse set of biophysically meaningful, often nonlocal couplings. As illustrated in Fig. 2, proximity-based edges are confined to local neighborhoods, whereas hydrogen-bond edges can span long sequence distances and link residues that are far apart in 3D but biochemically coupled. This highlights the key difference: SSProNet does not rely solely on arbitrary cutoffs but grounds its connectivity in physically interpretable interactions. The combined edge set feeds ProNet-style encoders (Wang et al., 2022a), ensuring that message passing operates on both generic spatial contacts and stabilizing biochemical interactions.

### 3.2 Node features augmented with biological priors

At each hierarchy level used by ProNet (Wang et al., 2022a) (residue, backbone, all-atom), SSProNet augments the SE(3)-invariant geometric descriptors $\mathcal{F}(G)_{\mathrm{base}}$, $\mathcal{F}(G)_{\mathrm{bb}}$, and $\mathcal{F}(G)_{\mathrm{all}}$ with two lightweight residue priors obtained from a standard annotator (e.g., DSSP (Hekkelman et al., 2025); see Appendix B): the secondary-structure label and solvent accessibility. These channels add interpretable biological context that complements ProNet's primarily local geometric descriptors.

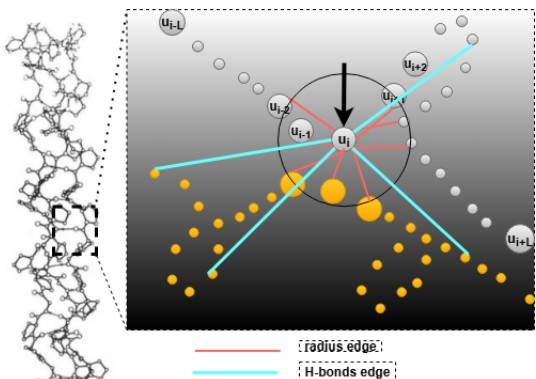

Figure 2: Comparison of graph construction strategies. Proximity-based graphs connect residues within a radius threshold, while SSProNet also adds hydrogen-bond edges that bridge distant sequence positions based on biophysical donor–acceptor rules. This expands the receptive field in a biologically meaningful way.

### 3.3 MODEL OVERVIEW

We retain ProNet's hierarchical encoder (Wang et al., 2022a) and change (i) the topology $E = E_{\mathrm{rad}} \cup E_{\mathrm{HB}}$ (see Section 3.1) and (ii) the node channels (see Section 3.2). Below we specify one interaction block; stacking $L$ blocks and adding a permutation-invariant readout completes SSProNet.

**Notation.** Let $\sigma(x) = x\,\mathrm{sigmoid}(x)$ (swish), $\odot$ denote the Hadamard product, and $\| \cdot \|$ denote vector concatenation. For node $i$, $\mathcal{N}(i) = \{j : (i, j) \in E\}$ is its neighbor set. The hidden width is $d \in \mathbb{N}$. For each edge $(i, j)$ we precompute three SE(3)-aware, ProNet-based descriptor families $\{\mathbf{f}_{ij}^{(k)}\}_{k=0}^2$ corresponding to: $k = 0$ (distance/angles), $k = 1$ (orientation or torsion; level-dependent), and $k = 2$ (positional).

**Edge gates.** We map descriptors to $d$-dimensional gates with small MLPs:

$$\mathbf{e}_{ij}^{(k)} = \phi_k\big(\mathbf{f}_{ij}^{(k)}\big) \in \mathbb{R}^d, \qquad k \in \{0, 1, 2\}. \tag{1}$$

where $\phi_k$ is a two-layer perceptron for stream $k$, and $\mathbf{e}_{ij}^{(k)}$ is the edge-wise gate that scales the message transmitted along $(i, j)$ in stream $k$.

**Interaction block (Hier-Geom-MP).** Given node states $\mathbf{x}^{(\ell)} = \{\mathbf{x}_i^{(\ell)}\}_{i \in V}$ at block $\ell$, we form a message view and a residual view:

$$\tilde{\mathbf{x}}_i^{(\ell)} = \sigma(\mathbf{A}\mathbf{x}_i^{(\ell)} + \mathbf{a}), \qquad \mathbf{r}_i^{(\ell)} = \sigma(\mathbf{B}\mathbf{x}_i^{(\ell)} + \mathbf{b}), \tag{2}$$

where $\mathbf{A}, \mathbf{B} \in \mathbb{R}^{d \times d}$ and $\mathbf{a}, \mathbf{b} \in \mathbb{R}^d$ are learnable; $\tilde{\mathbf{x}}_i^{(\ell)}$ is used to compute messages, while $\mathbf{r}_i^{(\ell)}$ provides the skip path.

Each stream $k$ applies an edge-gated GraphConv-style update (Morris et al., 2019):

$$\mathbf{m}_{ij}^{(k)} = \mathbf{e}_{ij}^{(k)} \odot \tilde{\mathbf{x}}_j^{(\ell)}, \qquad \textit{message sent from } j \textit{ to } i \textit{ in stream } k, \tag{3}$$

$$\mathbf{u}_i^{(k)} = \sum_{j \in \mathcal{N}(i)} \mathbf{m}_{ij}^{(k)}, \qquad \textit{neighbor aggregation at node } i, \tag{4}$$

$$\mathbf{h}_i^{(k)} = \sigma\big(\mathbf{L}^{(k)}\mathbf{u}_i^{(k)}\big), \qquad \textit{stream-specific linear head}, \tag{5}$$

where $\mathbf{L}^{(k)} \in \mathbb{R}^{d \times d}$ is learnable and has the same shape across streams.

**Fusion, mixing, and residual update.** We concatenate the three stream outputs, mix them with a small MLP, and add the residual view:

$$\mathbf{h}_i = \big\|_{k=0}^2 \mathbf{h}_i^{(k)} \in \mathbb{R}^{3d}, \qquad \mathbf{x}_i^{(\ell+1)} = \underbrace{\mathrm{MLP}\big(\mathbf{C}\,\mathbf{h}_i\big)}_{\text{stream mixing}} + \mathbf{r}_i^{(\ell)}. \tag{6}$$

where $\mathbf{C} \in \mathbb{R}^{d \times 3d}$ projects the concatenated streams back to width $d$, and MLP (2–3 layers with swish and dropout) mixes the streams before the skip addition.

**Readout and prediction.** After $L$ blocks, we pool node embeddings and predict task outputs:

$$\mathbf{h}_G \; = \; \sum_{i \in V} \mathbf{x}_i^{(L)}, \qquad \hat{y} \; = \; \mathrm{MLP}_{\mathrm{out}}(\mathbf{h}_G), \tag{7}$$

where the sum is permutation-invariant pooling over residues, and $\mathrm{MLP}_{\mathrm{out}}$ maps the graph embedding to logits (classification) or real values (regression).

**Summary.** Eqs. 1—6 define a ProNet-style Hier-Geom-MP block with three geometric streams and edge-gated messages; Eq. 7 is the permutation-invariant graph readout. SSProNet preserves these mechanics but *grounds $E$* in biophysics (radius scaffold + energy-filtered H-bonds) and *augments* node inputs with DSSP secondary-structure and solvent-accessibility priors.

## 4 EXPERIMENT

We evaluate our SSProNet on various protein tasks, including protein fold and reaction prediction, protein-ligand binding affinity prediction. Detailed descriptions of the datasets are provided in Section 4.1 . Detailed experimental setup and optimal hyperparameters are provided in Appendix A.

### 4.1 DATASETS

**Fold Dataset**. We use the same dataset as in (Wang et al., 2025; 2022a; Hou et al., 2018; Hermosilla et al., 2020). In total, this dataset contains 16,292 proteins from 1,195 folds. There are three test sets used to evaluate generalization ability:

- Fold: proteins from the same *superfamily* are unseen during training,
- Superfamily: proteins from the same *family* are unseen during training,
- Family: proteins from the same family are present during training.

Among the three test sets, **Fold** is the most challenging since it differs the most from the training distribution. In this task, 12,312 proteins are used for training, 736 for validation, 718 for Fold, 1,254 for Superfamily, and 1,272 for Family.

**Reaction Dataset**. For reaction classification, the 3D structures of 37,428 proteins representing 384 EC numbers are collected from the PDB database (Berman et al., 2000), and EC annotations for each protein are obtained from the SIFTS database (Dana et al., 2019). The dataset is split into 29,215 proteins for training, 2,562 for validation, and 5,651 for testing. Every EC number is represented in all three splits, and protein chains with more than 50% sequence similarity are grouped together.

**LBA Dataset**. Following (Somnath et al., 2021) and (Townshend et al., 2021), we perform ligand binding affinity (LBA) prediction on a subset of the commonly used PDBbind refined set (Wang et al., 2004; Liu et al., 2015). The curated dataset of 3,507 complexes is split into train/validation/test splits based on a 30% sequence identity threshold to evaluate model generalization on unseen proteins. For each protein–ligand complex, we predict the negative log-transformed binding affinity:

$$pK = -\log_{10}(K),$$

where $K$ is the binding constant measured in molar units.

### 4.2 BASELINES

Our main point of comparison is the recent state-of-the-art method **SCHull** (Wang et al., 2025), which currently leads performance on fold, reaction, and binding affinity tasks. To contextualize our contributions, we also benchmark SSProNet against a representative spectrum of methods in protein graph learning. Below we briefly describe each:

- **GCN** (Kipf, 2016): a classic semi-supervised GNN that propagates features layer by layer using a first-order spectral approximation.

Table 1: Accuracy (%) on protein fold and enzyme reaction classification tasks. *Avg. Time* is the average time per epoch (s). A dash (–) means not reported.

| Method | React | Avg. Time | Fold | | | | Avg. Time |
| --- | --- | --- | --- | --- | --- | --- | --- |
| | | | Fold | Super | Family | Avg. | |
| GCN (Kipf, 2016) | 67.3 | – | 16.8 | 21.3 | 82.8 | 40.3 | – |
| IEConv (Hermosilla et al., 2020) | 87.2 | – | 45.0 | 69.7 | 98.9 | 71.2 | – |
| DWNN (Li, 2022) | 76.7 | – | 31.8 | 37.8 | 85.2 | 51.5 | – |
| GearNet (Zhang et al., 2022) | 79.4 | – | 28.4 | 42.6 | 95.3 | 55.4 | – |
| HoloProt (Somnath et al., 2021) | 78.9 | – | – | – | – | – | – |
| MACE (Batatia et al., 2022) | – | – | 23.7 | 21.4 | 60.2 | 35.1 | 114 |
| MACE+SCHull (Wang et al., 2025) | – | – | 27.0 | 23.1 | 65.0 | 38.4 | 105 |
| SEGNN (Brandstetter et al., 2021) | – | – | 28.8 | 30.4 | 77.1 | 45.4 | 121 |
| SEGNN+SCHull (Wang et al., 2025) | – | – | 32.0 | 33.6 | 86.7 | 50.3 | 115 |
| GVP-GNN (Jing et al., 2020) | 65.5 | 320 | 16.0 | 22.5 | 83.8 | 40.8 | 106.3 |
| GVP-GNN + SCHull (Wang et al., 2025) | 77.1 | 345 | 24.5 | 27.1 | 88.0 | 47.1 | 111 |
| ProNet-Amino-Acid (Wang et al., 2022a) | 86.0 | 210 | 51.5 | 69.9 | 99.0 | 73.5 | 70.5 |
| ProNet-Amino-Acid+SCHull (Wang et al., 2025) | 87.9 | 221 | 52.2 | 73.9 | 99.2 | 75.1 | 69.3 |
| ProNet-Backbone (Wang et al., 2022a) | 86.4 | 213 | 52.7 | 70.3 | 99.3 | 74.1 | 74.1 |
| ProNet-Backbone+SCHull (Wang et al., 2025) | 88.1 | 230 | 56.1 | 74.6 | 99.4 | 76.7 | 75.8 |
| **SSProNet-Amino-Acid** (Ours) | 87.5 | 287 | 62.6 | 76.9 | 1.0 | 79.8 | 90.3 |
| **SSProNet-Backbone** (Ours) | **88.3** | 293 | **63.1** | **77.4** | 1.0 | **80.2** | 93.7 |

- **IEConv** (Hermosilla et al., 2020): uses a multi-graph representation combining structural connectivity and geometry, with a kernel that fuses intrinsic and extrinsic distances.

- **DWNN** (Li, 2022): an orientation-aware GNN with 3D directed weights, enabling explicit modeling of angular relations under equivariance.

- **GearNet** (Zhang et al., 2022): a geometry-aware residue graph encoder pretrained via contrastive and structural prediction tasks, which captures structural signals efficiently.

- **HoloProt** (Somnath et al., 2021): integrates surface geometry and residue topology in a multi-scale network, using superpixels to compress surface graphs and bridging layers in message passing.

- **MACE** (Batatia et al., 2022): supports higher-order message passing (beyond pairwise) in an equivariant framework, reducing the depth required while retaining expressivity.

- **SEGNN** (Brandstetter et al., 2021): extends E(3) equivariant GNNs by allowing steerable node and edge features, processed by nonlinear steerable MLPs with tensorial combinations.

- **GVP-GNN** (Jing et al., 2020): replaces standard MLPs with Geometric Vector Perceptrons that jointly handle invariant scalars and equivariant vectors, enabling richer geometric reasoning.

- **ProNet** (Wang et al., 2022a): a hierarchical 3D graph architecture for proteins that ensures completeness across amino acid, backbone, and all-atom levels. It employs hierarchical message propagation (Hier-Geom-MP) for flexible traversal across granularities.

### 4.3 TASK 1: FOLD CLASSIFICATION

Protein fold classification (Hou et al., 2018; Levitt & Chothia, 1976) is a fundamental task for understanding protein structure–function relationships and evolutionary patterns. Following the dataset and experimental setup of (Wang et al., 2025), we evaluate our methods on this task. A detailed description of the dataset is provided in Appendix 4.1. In total, the dataset comprises 16,712 proteins spanning 1,195 folds. It includes three test sets: Fold, Superfamily, and Family. We report the accuracies on each of these test sets, as well as their average, in Table 1. In line with (Wang et al., 2025), to examine how SSProNet facilitates the capture of global structural information, each test set is further divided into four subsets based on graph size, with node counts capped at 150, 300, 450, and 600.

Table 1 demonstrates that on the FOLD dataset, SSProNet achieves the best accuracy on Fold/Superfamily/Family (63.10/77.42/100.0) and the highest average (80.17), surpassing the SCHull-based baselines (Wang et al., 2025) by +7.0, +2.82, +0.6, and +3.47 points, respectively. This comes with a ∼24–27% increase in per-epoch training time.

## 4.4 TASK 2: REACTION CLASSIFICATION

Enzymes are proteins that act as biological catalysts, and their functions are systematically classified by enzyme commission (EC) numbers, which group enzymes according to the reactions they catalyze (Omelchenko et al., 2010). In this experiment, we assess the SSProNet model on the reaction classification task, using the same dataset and experimental setup as described in (Wang et al., 2025; 2022a). Further details on the dataset and the training, validation, and test splits are provided in Appendix 4.1.

For the EC dataset, Table 1 shows that SSProNet-Backbone establishes a new state of the art, achieving the highest accuracy (88.3%) and surpassing the previous best ProNet-Backbone+SCHull baseline (88.1%) (Wang et al., 2025). This gain, although modest in absolute terms, confirms that our secondary-structure–aware design improves generalization beyond existing methods. The improvement comes at the cost of a moderate increase in runtime (about 27–35% per epoch).

## 4.5 TASK 3: LIGAND BINDING AFFINITY

Predicting protein–ligand binding affinity (LBA) is a fundamental task in drug discovery, with direct impact on downstream applications such as virtual screening and lead optimization. For this task, we adopt our integrated SSProNet model to predict LBA. The dataset is derived from PDB-bind (Wang et al., 2004; Liu et al., 2015) following the experimental protocols outlined in (Wang et al., 2025; Jing et al., 2020), and we use the default dataset split (see Appendix 4.1 for details). Evaluation is conducted using multiple statistical metrics—RMSE, Pearson, Spearman, and Kendall correlations—to assess how SSProNet improves the learning capacity and generalization of GNN-based models.

Table 2: RMSE/Pearson Correlation/Spearman Correlation/Kendall Correlation on the LBA test set. *Avg. Time* is the average running time per epoch. Arrows indicate whether lower or higher is better. A dash (−) means not reported.

| Method | LBA | | | | Avg. Time |
|---|---|---|---|---|---|
| | RMSE (↓) | Pearson (↑) | Spearman (↑) | Kendall (↑) | |
| IEConv (Hermosilla et al., 2020) | 1.554 | 0.414 | 0.428 | – | – |
| HoloProt-Full Surface (Somnath et al., 2021) | 1.464 | 0.509 | 0.500 | – | – |
| HoloProt-Superpixel (Somnath et al., 2021) | 1.491 | 0.491 | 0.482 | – | – |
| GVP-GNN (Jing et al., 2020) | 1.529 | 0.441 | 0.432 | 0.301 | 48.6 |
| GVP-GNN + SCHull (Wang et al., 2025) | 1.401 | 0.475 | 0.459 | 0.335 | 53.6 |
| ProNet-Amino–Acid (Wang et al., 2022a) | 1.455 | 0.536 | 0.526 | 0.465 | 31.7 |
| ProNet-Amino–Acid+SCHull (Wang et al., 2025) | 1.355 | 0.556 | 0.568 | 0.512 | 33.9 |
| ProNet-Backbone (Wang et al., 2022a) | 1.458 | 0.546 | 0.550 | 0.481 | 32.1 |
| ProNet-Backbone+SCHull (Wang et al., 2025) | **1.321** | 0.581 | 0.578 | **0.535** | 34.4 |
| **SSProNet-Amino-Acid** (Ours) | 1.354 | 0.607 | 0.601 | 0.487 | 47.3 |
| **SSProNet-Backbone** (Ours) | 1.382 | **0.613** | **0.616** | 0.498 | 54.7 |

As shown in Table 2, our SSProNet model establishes a new state of the art on the LBA benchmark. In particular, SSProNet–Backbone achieves the highest correlation scores (Pearson = **0.613**, Spearman = **0.616**), surpassing the strongest SCHull (Wang et al., 2025) baseline by +0.032 and +0.038, respectively. Although RMSE (1.382 vs. 1.321) and Kendall (0.498 vs. 0.535) remain slightly below the best baseline, the improvements in correlation metrics are significant, demonstrating the strength of our secondary-structure–aware design. These gains come with a moderate increase in training time (54.7 s vs. 34.4 s per epoch, see Table 2).

## 4.6 ABLATION STUDIES

To better understand the contribution of individual design choices in SSProNet, we conduct ablation experiments on the LBA dataset using the amino acid representation.

**Influence of the energy threshold.** As shown in Table 3, the choice of energy cutoff substantially influences LBA performance. The most permissive threshold (−0.1 kcal/mol), which retains both strong and weak hydrogen bonds, achieves the best overall results: RMSE = 1.336, Pearson = 0.612, Spearman = 0.609, Kendall = 0.432. This suggests that weak hydrogen bonds still carry useful geometric and interaction information that benefits predictive accuracy when included in the graph.

Table 3: Ablation study on the influence of the energy threshold for constructing H-bond edges. Results are reported on the LBA test set. Arrows indicate whether lower or higher is better.

| Energy Threshold | LBA | | | | Avg. Time (s) |
|---|---|---|---|---|---|
| | RMSE ($\downarrow$) | Pearson ($\uparrow$) | Spearman ($\uparrow$) | Kendall ($\uparrow$) | |
| -0.1 | **1.336** | **0.612** | **0.609** | **0.432** | 57.3 |
| -1.5 | 1.354 | 0.605 | 0.601 | 0.424 | 47.9 |
| -2.5 | 1.354 | 0.607 | 0.601 | 0.424 | 45.8 |
| -3.5 | 1.349 | 0.605 | 0.601 | 0.426 | 43.7 |

By contrast, more stringent thresholds ($-1.5$ to $-3.5$ kcal/mol) progressively exclude weaker bonds and lead to sparser graphs. While this reduces runtime (from $57.3$ s at $-0.1$ to $43.7$ s at $-3.5$ per epoch), it also slightly diminishes correlation metrics (Pearson $\approx 0.605$, Spearman $\approx 0.601$, Kendall $\approx 0.424$–$0.426$). The results therefore reveal a clear trade-off: keeping weaker bonds improves the model's ability to capture global affinity trends, whereas filtering them out yields time efficiency gains but weaker predictive consistency.

Compared to Table 2, the $-0.1$ setting surpasses our default SSProNet–Amino-Acid model on RMSE ($1.336$ vs. $1.354$) and correlations (Pearson $0.612$ vs. $0.607$, Spearman $0.609$ vs. $0.601$), although Kendall correlation drops ($0.432$ vs. $0.487$). Relative to the strongest SCHull (Wang et al., 2025) baseline, our ablation improves Pearson/Spearman but remains slightly behind in RMSE and Kendall.

### 4.7 Effect of graph topology and DSSP-derived features

Table 4: Ablation study on the effect of removing different information sources. Results are reported on the LBA test set (best epoch). Arrows indicate whether lower or higher is better.

| Ablation | LBA | | | |
|---|---|---|---|---|
| | RMSE ($\downarrow$) | Pearson ($\uparrow$) | Spearman ($\uparrow$) | Kendall ($\uparrow$) |
| Removing Radius Edges (H-bond only) | 1.385 | 0.570 | 0.579 | 0.408 |
| Removing SS (keep ACC) | 1.361 | 0.606 | 0.599 | 0.424 |
| Removing ACC (keep SS) | **1.362** | **0.611** | **0.606** | **0.429** |

Table 4 shows three ablations. First, removing the radius graph and keeping only hydrogen-bond edges degrades performance (RMSE $= 1.385$, Pearson $= 0.570$), indicating that proximity-based edges provide complementary structural context beyond H-bond connectivity.

Second, comparing the two node-annotation ablations reveals that *secondary structure (SS) is more useful than solvent accessibility (ACC)* for LBA. When we *remove ACC* but keep SS, we obtain the strongest correlations (Pearson $= 0.611$, Spearman $= 0.606$, Kendall $= 0.429$) with virtually the same RMSE as the w/o-SS variant ($1.362$ vs. $1.361$). Conversely, *removing SS* yields slightly lower correlations (Pearson $= 0.606$, Spearman $= 0.599$, Kendall $= 0.424$). Taken together, these results suggest: (i) radius edges complement H-bonds and should be retained; (ii) SS carries the dominant DSSP signal for affinity prediction, while ACC contributes less.

## 5 Conclusion

We introduce SSProNet, a new graph neural network model that enriches node features with per-residue secondary-structure labels and adds hydrogen-bond edges on top of regular proximity based edges. Across Fold, Reaction, and LBA benchmarks, our model yields competitive and improved performance. Ablations identify the main drivers: radius-based proximity edges are indispensable for affinity prediction; secondary-structure cues contribute more than solvent accessibility; and permissive H-bond thresholds that retain weaker bonds modestly improve generalization at a runtime cost. Overall, grounding protein graphs in biophysical interactions provides an effective inductive bias, improving both accuracy and interpretability.

## REPRODUCIBILITY STATEMENT

In this paper, we have provided implementation details in Section 4.1 and Appendix A. We will provide the code upon request during the review process and promise to release the code upon the paper's publication.

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

## A  Hyperparameter Details & Experimental Setup

This section describes the full experiment setup for each task considered in this paper. The implementation of our methods is based on the PyTorch (Paszke et al., 2019) and Pytorch Geometric (Fey & Lenssen, 2019), and all models are trained with the Adam optimizer (Kingma & Ba, 2015). All experiments are conducted on a single NVIDIA Tesla V100 32GB GPU. The search space for model and training hyperparameters are listed in Table 5. Note that we select hyperparameters at the amino acid and backbone levels by the same search space, and optimal hyperparameters are chosen by the performance on the validation set.

Table 5: Model and training hyperparameters for protein-related datasets.

| Hyperparameter | React | Fold | LBA |
|---|---|---|---|
| Number of layers | 3, 4, 5 | 3, 4, 5 | 3, 4, 5 |
| Hidden channels | 64, 128, 256 | 64, 128, 256 | 128, 192, 256 |
| Cutoff | 6, 8, 10 | 6, 8, 10 | 6, 8, 10 |
| Dropout | 0.2, 0.3, 0.5 | 0.2, 0.3, 0.5 | 0.2, 0.3 |
| Epochs | 500, 1000 | 500, 1000 | 500, 800 |
| Batch size | 16, 32 | 16, 32 | 8, 16, 32 |
| Learning rate | 1e-4, 5e-4 | 1e-4, 5e-4 | 5e-5, 1e-4, 2e-4 |
| Learning rate scheduler | step_lr | step_lr | step_lr |
| Learning rate decay factor | 0.5 | 0.5 | 0.5 |
| Learning rate decay epochs | 50, 100 ,150 | 100, 150, 200 | 50, 100 , 150 |

## B  DSSP Preprocessing and Integration

**Role of DSSP and how we use it.**  The Dictionary of Secondary Structure of Proteins (DSSP) is a long-standing standard for deriving residue-level annotations (secondary structure, hydrogen bonds, solvent accessibility, backbone geometry) directly from 3D coordinates (Hekkelman et al., 2025). In our pipeline we install DSSP locally (version 2.3.0) and use it to annotate each protein chain, then feed those annotations into our graph construction and node features. Concretely, for each residue we use: (1) the *primary secondary-structure code* (H, E, T, S, G, B, I; defaults to coil if unassigned), (2) the *solvent-accessible surface area* (ACC), (3) backbone dihedrals (PHI, PSI), and (4) *hydrogen-bond partners with energies*. These DSSP attributes allow us to complement purely geometric proximity with biochemical constraints (e.g., hydrogen bonds) and physically meaningful local context (ACC, dihedrals).

**Example of produced `.dssp` output.**  Below is a short excerpt from one of our generated DSSP files (1b6v.A.dssp); columns are truncated for readability but show the key fields we use:

```
#  RESIDUE AA STRUCTURE BP1 BP2  ACC     N-H-->O    O-->H-N   ...  PHI   PSI   ...
13   12 A M  B  <  +a  47  0A   9    -4,-2.3    2,-0.2   ... -100.3 123.4 ...
14   13 A D       +     0   0  11    33,-2.7    3,-0.2   ... -135.7  79.8 ...
19   18 A A  S    S-    0   0  39    61,-0.0    2,-0.7   ... -137.7 146.5 ...
22   21 A S  S >  S-    0   0  77     1,-0.1    3,-2.0   ...   72.1 115.0 ...
24   23 A N  T 3> +     0   0  75     1,-0.1    4,-2.4   ... -100.8   6.1 ...
```

Each residue line includes: (i) indices and chain ID, (ii) amino-acid code (AA), (iii) the STRUCTURE symbol (e.g., H helix, E strand, T turn, S bend), (iv) ACC (solvent accessibility), (v) hydrogen-bond partners and energies for N–H→O and O→H–N (pairs like offset,energy), and (vi) backbone geometry (PHI, PSI).

**How we use these fields in our model.**  We parse the produced .dssp files and attach their information to each residue/node of the protein graph. Secondary structure is mapped to an 8-way categorical label and one-hot encoded; ACC is kept as a scalar feature. Hydrogen-bond partners are

converted into additional *edges* in the graph: for each residue, we add edges to the residues indicated by DSSP's H-bond partners (using the provided offsets), optionally filtering by bond energy (more negative indicates stronger bonding). These DSSP-derived edges are merged with the usual radius-based proximity edges, duplicates are removed, and self-loops are dropped. On the node side, SS and ACC are concatenated with the sequence/structure features used by SSProNet (amino-acid one-hot; and, depending on the chosen level, backbone and/or side-chain embeddings). This way, the model simultaneously "sees" short-range geometric contacts and longer-range biochemical links, improving its capacity to capture secondary-structure regularities and nonlocal constraints (e.g., $\beta$-sheet hydrogen-bonding).

## C LLM USAGE DISCLOSURE

During the paper writing process, the authors utilize LLMs as tools to formalize word choice and correct grammatical mistakes.

