# OpenReview forum: "SSProNet: Secondary Structure aware Graph Neural Network for Protein Representation Learning"
_ICLR.cc/2026/Conference — ICLR 2026 Conference Withdrawn Submission_

### Official Review · Reviewer_G4QC · 2025-10-28

**Soundness:** 1
**Presentation:** 2
**Contribution:** 1
**Rating:** 2
**Confidence:** 4

**Summary:**

This paper proposed a method to construct graphs for learning on protein structures. It adds secondary structure information to node features and hydrogen bonding information to edge features.

**Strengths:**

- Incorporating secondary structure and hydrogen bonding information is reasonable.
- The paper is overall organized and easy to follow.

**Weaknesses:**

- The main contribution of this paper is mostly feature engineering. However, neither secondary structure or hydrogen bonding is novel features or features that have been less investigated. The graph neural network architecture is standard. There is little methodological innovation or new insights in this work.
- The motivation of using secondary structure and hbond information is too general. It is unclear  it is unclear what improvements these features are expected to bring and how these features would improve. The evaluation benchmarks are general.
- The performance gains on the three benchmark tasks are marginal.

**Questions:**

See weaknesses

---

### Official Review · Reviewer_m7T4 · 2025-10-28

**Soundness:** 3
**Presentation:** 1
**Contribution:** 2
**Rating:** 2
**Confidence:** 4

**Summary:**

This paper proposes SSProNet, a GNN for learning protein representations. The authors argue that standard graph-building methods, which just connect nearby atoms, are not very meaningful. Instead, SSProNet uses a more biologically-informed approach based on protein secondary structure. The model then uses the existing ProNet GNN architecture to process this new, augmented graph. The authors show this method improves performance on tasks like fold classification and ligand binding.

**Strengths:**

1. The core idea is chemically sound. Using hydrogen bonds to define the graph is a more physical and meaningful way to represent protein structure than just using an arbitrary distance cutoff.

2. The paper does a good job of isolating its contribution.  It re-uses a strong, existing GNN architecture and only changes the input graph and node features.  This clearly shows the benefit of the new, biology-aware graph.

**Weaknesses:**

1. The method is an incremental improvement. The GNN architecture itself is taken directly from ProNet. The novelty is in the pre-processing, not in the learning model.

2. This method adds a required pre-processing step: running DSSP on every protein to find all the hydrogen bonds and secondary structures before training can even start.

3. The new graph structure significantly increases training time.

4. Performance improvement is not consistant. On the LBA task, the model is actually worse than the baseline on two of the four metrics.

5. H-bond edges makes performance much worse.  This means the H-bonds are just a supplement, not a replacement for standard proximity graphs.

6. The abstract and introduction claim the model uses H-bonds weighted by their energetic strength. However, the method section states that the energies are used for filtering the edges rather than as per-edge weights. This is a contradiction and makes the energy-weighted claim misleading.

**Questions:**

See weakness part.

---

### Official Review · Reviewer_snsM · 2025-10-31

**Soundness:** 3
**Presentation:** 3
**Contribution:** 2
**Rating:** 4
**Confidence:** 3

**Summary:**

The paper proposes a graph neural network for protein learning that addresses the secondary structures. Key to the method is the introduction of the hydrogen-bond edges that address the biophysical structures of proteins, in addition to the proximity edges that focus merely on the geometric structures. It further introduces some network architectural designs based on this. Experiments show the effectiveness of the new network architecture on various protein learning tasks.

**Strengths:**

- The work is well-motivated by analyzing the nature of the protein learning problems. The proposed network addresses the biophysical structures in these problems, in addition to the geometric features.
- The paper proposes a valid framework following this intuition with performance improvements.

**Weaknesses:**

- I am not from a biology background, but I wonder, for the three tasks in Section 4.3-4.5, how closely are they related to the hydrogen-bond information? To my feeling, the network achieves better performances not only because of a better architectural design, but it also relies on the additional information of hydrogen-bond edges, which essentially comes from additional annotations (e.g., from DSSP). In this sense, the network actually takes more inputs than the baseline methods, and this is required at both training and inference time?

- If the previous point is true (that the network actually relies on more inputs/annotations), the performance improvements compared to the baselines look a bit marginal? -- correct me if the gap is actually quite large, as I'm not familiar with these tasks.

**Questions:**

- See weaknesses. I'm mostly interested in: (i) if additional inputs are introduced, and (ii) if so, how relevant are these inputs to the tasks.
- The two types of edges $E_{\text{rad}}$ and $E_{\text{HB}}$ are treated completely equally?

---

### Official Review · Reviewer_1yVb · 2025-10-31

**Soundness:** 3
**Presentation:** 2
**Contribution:** 2
**Rating:** 2
**Confidence:** 3

**Summary:**

This paper proposes SSProNet, a graph neural network for protein representation learning that incorporates secondary structure as additional node features and hydrogen bond awareness as additional weighted edges. The authors claim that their model achieves state of the art results on protein fold classification, enzyme reaction classification, and ligand binding affinity.

**Strengths:**

- I like the motivation of incorporating domain knowledge to increase model interpretability. Considering hydrogen bonds is a nice way of including longer-range interactions in an informed manner.
- In general, the paper is quite clearly written and easy to follow.

**Weaknesses:**

- My main concern is with the novelty of this work, since most of the performance gain seems to come from enriching node features with secondary structure information. While this is a nice practical result, I’m not sure that it constitutes enough of a machine learning contribution, since there are several other models that improve model performance by injecting structural information [1-3], and are much more general in their results.
    - Can the authors please clarify how their model meaningfully builds upon these works?
- I’m not entirely convinced by the experimental results in this work either. In particular:
    - The authors are missing baseline results from models that are better designed for long-range reasoning, which their H-bond edge construction seeks to tackle. An analysis comparing SSProNet to leading graph transformer or GNN with virtual node models would provide a more fair comparison here and strengthen their claim.
    - The results for LBA are not very strong, given that ProNet-Backbone+SCHull outperforms SSProNet on half of the evaluation metrics, and is over 30% faster.
    - Are there any additional datasets where the authors can show stronger results?

Overall, I think this work would be more fitting as a workshop paper or for a more applied venue. Therefore, I recommend rejection.

References:
1. Bouritas et. al. Improving Graph Neural Network Expressivity via Subgraph Isomorphism Counting. In ICLR, 2021.
2. Barcelo et. al. Graph Neural Networks with Local Graph Parameters. In NeurIPS, 2021.
3. Jin et al. Homomorphism Counts for Graph Neural Networks. In ICML, 2024.

**Questions:**

See weaknesses.

---

### Note · Authors · 2026-01-21

I have read and agree with the venue's withdrawal policy on behalf of myself and my co-authors.